# Functional Analysis of the PoSERK-Interacting Protein PorbcL in the Embryogenic Callus Formation of Tree Peony (*Paeonia ostii* T. Hong et J. X. Zhang)

**DOI:** 10.3390/plants13192697

**Published:** 2024-09-26

**Authors:** Yinglong Song, Jiange Wang, Jiale Zhu, Wenqian Shang, Wenqing Jia, Yuke Sun, Songlin He, Xitian Yang, Zheng Wang

**Affiliations:** 1Postdoctoral Innovation Practice Base, Henan Institute of Science and Technology, Xinxiang 453003, China; edward_song1989@163.com; 2Postdoctoral Workstation, Henan Bainong Seed Industry Co., Ltd., Xinxiang 453003, China; 3College of Landscape Architecture and Art, Henan Agricultural University, Zhengzhou 450002, China; lucky_jiangew@163.com (J.W.); le95952076@163.com (J.Z.); qianqian656@163.com (W.S.); yukesun1@163.com (Y.S.); 4College of Horticulture and Landscape Architecture, Henan Institute of Science and Technology, Xinxiang 453003, China; jiawq2022@hist.edu.cn

**Keywords:** *Paeonia ostii*, *PorbcL*, protein interaction, regulatory mechanism, somatic embryogenesis

## Abstract

*SERK* is a marker gene for early somatic embryogenesis. We screened and functionally verified a SERK-interacting protein to gain insights into tree-peony somatic embryogenesis. Using PoSERK as bait, we identified PorbcL (i.e., the large subunit of Rubisco) as a SERK-interacting protein from a yeast two-hybrid (Y2H) library of cDNA from developing tree-peony somatic embryos. The interaction between PorbcL and PoSERK was verified by Y2H and bimolecular fluorescence complementation analyses. *PorbcL* encodes a 586-amino-acid acidic non-secreted hydrophobic non-transmembrane protein that is mainly localized in the chloroplast and plasma membrane. *PorbcL* was highly expressed in tree-peony roots and flowers and was up-regulated during zygotic embryo development. *PorbcL* overexpression caused the up-regulation of *PoSERK* (encoding somatic embryogenesis receptor-like kinase), *PoAGL15* (encoding agamous-like 15), and *PoGPT1* (encoding glucose-6-phosphate translocator), while it caused the down-regulation of *PoLEC1* (encoding leafy cotyledon 1) in tree-peony callus. *PorbcL* overexpression led to increased indole-3-acetic acid (IAA) content but decreasing contents of abscisic acid (ABA) and 6-benzyladenosine (BAPR). The changes in gene expression, high IAA levels, and increased ratio of IAA to ABA, BAPR, 1-Aminocyclopropanecarboxylic acid (ACC), 5-Deoxystrigol (5DS), and brassinolide (BL) promoted embryogenesis. These results provide a foundation for establishing a tree-peony embryogenic callus system.

## 1. Introduction

Tree peony (*Paeonia suffruticosa* Andr.) is a perennial deciduous small shrub in the family Paeoniaceae and genus *Paeonia* [1,2]. It is a world-renowned flower and a traditional characteristic flower of China [3,4]. It has a long history of cultivation and an important cultural heritage, and it is highly valued as an ornamental, medicinal, and oil-rich plant resource [1,5,6,7]. Since ancient times, it has been revered as the “king of flowers” [8,9]. It occupies an important position in the domestic and international flower market due to its significant economic, scientific, and cultural values.

The development of the flower industry is ultimately constrained by the richness and quality of germplasm resources [10,11,12]. Over more than 1000 years of breeding, through selection and hybridization, the number and diversity of peony resources have increased. However, overall progress has been slow, with only a limited number of tree-peony resources being utilized and popularized [13,14,15]. The existing varieties lack diversity and have a narrow genetic basis. Most new varieties are relatively uniform in color, flower type, and fragrance. There is a lack of yellow or orange hues as well as long flowering periods, extended flower branches, strong stress resistance, and various potted-plant types, all of which are urgently needed for the development of the tree peony industry [16,17,18]. To address these limitations, it is necessary to find new technologies and methods to widen the range of germplasm resources and cultivate new, high-quality varieties with enhanced resistance [18,19]. Molecular breeding offers potential solutions by targeting single or multiple regulatory genes associated with specific traits for accurate trait selection in offspring [17,20]. Hence, it may be one of the most effective approaches to overcome developmental constraints within the tree peony industry. Because this breeding method relies on a mature tissue culture-regeneration system and better maintain genetic stability, it is crucial to construct a somatic embryo-regeneration system for tree peony.

The tree-peony tissue culture-regeneration system needs further improvement [8,21]. Previous research on tree-peony tissue cultures has focused on the selection of explants, the type and formula of the culture medium, the culture conditions, phytohormones, and other exogenous additives [22,23,24]. In the existing regeneration systems, seedlings form directly from explants. There have been few reports on the construction of a somatic embryo-regeneration system [25,26]. The main reasons for this are that it is difficult to induce the formation of embryonic callus tissue from the non-embryonic callus tissue of tree peony, and there is a low frequency of embryonic cell types, such as spherical and heart-shaped embryos, in callus tissue [9,21]. The process of plant embryogenesis is regulated by various factors such as the types and concentrations of phytohormones, the expression levels of particular genes, and environmental factors [27,28,29,30]. It is also controlled by key regulatory transcription factors and embryogenesis-related functional enzymes, such as *SERK*, *LEC*, *AGL15*, *GPT1*, and *RbcL* [9,31,32].

Ribulose-1,5-bisphosphate carboxylase/oxygenase (Rubisco) is present in the chloroplasts of all plants and consists of eight large subunits (rbcL) and eight small subunits (rbcS). It is a key enzyme that determines the rate of carbon assimilation in photosynthesis, and it is closely related to growth and development and the accumulation of secondary metabolites [33,34,35]. Because Rubisco’s catalytic sites are mainly located in the rbcL subunit, research on the structure and function of this component is particularly important [36,37]. Functional analyses of rbcL can provide a new research approach and direction for improving plant photosynthetic metabolism and morphogenesis [38]. Previous studies have shown that the *rbcL* gene responds to, and participates in, the plant response to abiotic stress [32,39]. A previous study detected a direct correlation between the transcript level of *CorbcL* and the oil and seed yields of *Camellia oleifera* [40]. In castor bean, RcrbcL was found to interact with RcGST F11 to regulate the sugar metabolism pathway, thereby regulating plant growth and development [41]. However, no studies have yet explored whether rbcL plays a regulatory role in embryogenesis.

In this study, PoSERK was employed as bait to screen a yeast two-hybrid (Y2H) library constructed using cDNA from developing tree-peony embryos, and a positive interacting partner, the large subunit of ribulose-1,5-bisphosphate carboxylase/oxygenase (Rubisco), PorbcL, was identified. The interaction between PorbcL and PoSERK was further confirmed using Y2H and bimolecular fluorescence complementation (BiFC) analyses. The function of PorbcL during the somatic embryogenesis of tree peony was further analyzed. The expression levels of *PorbcL* were detected in different organs and during the induction of embryonic callus formation in tree peony. The effects of *PorbcL* on the transcription levels of other key genes involved in somatic embryogenesis, including *PoSERK*, *PoLEC1*, and *PoAGL15*, as well as its regulatory role in endogenous hormone content changes, were analyzed to elucidate the function of PorbcL during somatic embryogenesis in tree peony. This research will provide new insights into the role of *PorbcL* in the induction of embryogenic callus and the construction of a somatic embryo-regeneration system in tree peony.

## 2. Results

### 2.1. Construction of Yeast Library and Screening of PoSERK-Interacting Proteins

As shown in Figure 1, the concentration of total RNA extracted from the tree-peony embryos was 2709 ng/μL, and it had an OD260/280 ratio of 2.02 (Appendix A). Distinct 18S and 28S bands were detected in the electrophoretogram, indicating the good purity and concentration of the extracted RNA (Figure 1A). The quality assessment of the isolated and purified mRNA revealed an OD260/OD280 value of 1.92 (Figure 1B). The gel electrophoresis results revealed an mRNA size distribution of 1000 to 5000 bp, indicating high quality and purity. The primary library had a capacity of 3.9 × 10^6^ cfu/mL, with a total clone count of 7.8 × 10^6^ cfu. Twenty-four randomly selected single colonies were subjected to PCR detection, and most of the bands were concentrated around 500 bp (Figure 1C,D). The secondary library had a capacity of 6.2 × 10^6^ cfu/mL, with a total clone count of 1.2 × 10^7^ cfu. PCR detection was performed on 24 randomly selected single colonies, and the bands were mostly concentrated between 500 bp and 750 bp (Figure 1E,F). These results confirmed that the quality of the libraries met the requirements for subsequent screening.

The pBT3-STE-PoSERK and secondary library plasmids were co-transformed into NMY51-competent cells for library screening. The cells were sequentially plated on SD-TL, SD-TLHA, and SD-TLHA+X-α-gal plates. Positive single colonies were picked for PCR identification, and the PCR products were sequenced by the Sangon Biotech Co., Ltd. (Shanghai, China). Through Blast comparison and analysis, a total of 20 interactor sequences were obtained (Table 1 and Appendix A). These sequences encoded the large subunit of Rubisco (rbcL), glucose-6-phosphate/phosphate translocator (GPT1), GPR107-like protein, post-GPI attachment to proteins factor 3 (PGAP3), Kelch repeat-containing protein At3g27220-like, transcriptional adapter ADA2-like, Golgin candidate 4, probable cysteine protease RD21B, endoplasmin homolog, 60S ribosomal protein L4, FRIGIDA-like protein 4a, uncharacterized GPI-anchored protein At4g28100, signal peptide peptidase, probable acyl-activating enzyme 17, tetraspanin-8, universal stress protein A-like protein, tubulin beta-1 chain, transcription factor bHLH35, NADPH-dependent aldo, and heat shock protein 90-5. These proteins play crucial roles in various physiological and biochemical processes such as photosynthetic metabolism, embryo development, auxin and cytokinin regulation, and the regulation of cell differentiation and flower development. These processes hold significant research value.

### 2.2. Bioinformatics Analysis of PorbcL

*PorbcL* was selected for bioinformatic analysis. Its ORF spans 1761 bp in length, encoding a polypeptide consisting of 586 amino acids, with a predicted molecular weight of 61.98 kDa. The amino acid sequence of this putative protein incorporates a highly conserved GroEL_like type I chaperonin superfamily domain (Figure 2A,B).

The online prediction analysis revealed that PorbcL lacks a signal peptide, indicating its non-secretory nature (Figure 2C). The transmembrane domain prediction analysis revealed that PorbcL lacks a transmembrane domain and was, thus, not a transmembrane protein (Figure 2D). The physicochemical properties of the predicted PorbcL protein were analyzed using online software; the protein instability index of 28.69, fat solubility index of 104.44, and total average hydrophilicity index of −0.023 indicated that PorbcL was a stable hydrophobic liposoluble protein (Appendix A). The predicted isoelectric point of the protein was 5.15. The protein contained more acidic amino acid residues than basic residues; thus, PorbcL is classified as an acidic protein.

The PorbcL lacked a signal peptide, which suggests its status as a non-secretory protein (Figure 2C). Furthermore, the transmembrane domain prediction analysis confirmed that PorbcL does not possess a transmembrane domain, hence classifying it as non-transmembrane (Figure 2D). Analysis of the physicochemical properties of the predicted PorbcL protein, using online software, revealed an instability index of 28.69, a fat solubility index of 104.44, and a total average hydrophilicity index of −0.023, all of which point towards PorbcL being a stable, hydrophobic, and liposoluble protein. The predicted isoelectric point of the protein was 5.15, and its acidic amino acid residues exceeded those of basic residues, classifying it as an acidic protein.

It exhibited a high degree of homology with its counterparts found in *Quercus lobata*, *Arabidopsis thaliana*, *Hevea brasiliensis*, *Populus trichocarpa*, and *Vitis riparia*, resulting in the gene being designated as *PorbcL* (Figure 2B,E).

### 2.3. Subcellular Localization of PorbcL

The coding sequence of PorbcL was integrated into a GFP-tagged vector, which was subsequently transfected into the lower epidermal cells of *N. benthamiana* leaves to elucidate its subcellular localization. As shown in Figure 3, green fluorescence signals from the GFP-PorbcL protein were observed in the chloroplast and plasma membrane of tobacco leaf cells. The GFP signal did not overlap with the DAPI staining signal (i.e., blue fluorescence) in the nucleus. The confirmation of these results indicated that the PorbcL protein was localized to both the chloroplast and the plasma membrane.

### 2.4. Verification of the Interaction between PoSERK and PorbcL

To confirm the physical interaction between PoSERK and PorbcL, we co-transformed pBT3-STE-PoSERK and pBT3-N-PorbcL into yeast NMY51-competent cells. Following plating on SD-TL and SD-TLHA agar plates, colonies exhibited normal growth, indicating successful co-transformation (Figure 4A,B). Positive single colonies were picked and diluted 10-, 10^2^-, 10^3^-, and 10^4^-fold, and each dilution was spotted on SD-TLHA+X-α-gal plates (Figure 4C). Normal colony growth was observed, and the colonies turned blue, confirming the interaction between PoSERK and PorbcL.

The YFP-N-PoSERK and YFP-C-PorbcL plasmids were separately transformed into *A. tumefaciens* strain GV3101. Subsequently, bacterial suspensions of the two strains were mixed in equal proportions and used to infect the lower epidermal cells of *N. benthamiana* leaves. After 3 days of incubation, laser scanning confocal microscopy was employed to observe the distinct interaction between PoSERK and PorbcL. Fluorescence signals were detected in both the cell membrane and chloroplasts (Figure 4D–G), demonstrating that the interaction took place at both the cell membrane and within the chloroplasts.

### 2.5. Expression Pattern Analysis of PorbcL

Using RT-qPCR analysis, this study aimed to characterize the spatiotemporal expression patterns of *PorbcL* and quantify its transcript abundance across tree-peony organs and at various stages of embryonic development. This was achieved by synthesizing cDNAs from RNA extracts of roots, stems, leaves, flowers, seeds, and pods from various developmental stages. The organs were subsequently ranked in descending order according to *PorbcL* transcript levels, revealing a hierarchy: flower > root > leaf > stem > seed (Figure 5A). The abundance of *PorbcL* transcripts gradually decreased from 5 d to 60 d of zygotic embryo development but then significantly increased during the later stage, from 60 d to 110 d (Figure 5B).

### 2.6. RT-qPCR Analysis of Genes Expression Levels

The *PorbcL* overexpression vector, pCAMBIA1302-*PorbcL*, was successfully introduced into tree-peony calli via *Agrobacterium*-mediated transformation. The initial detection revealed low but comparable *PorbcL* transcript levels in both WT and CK calli, with no statistically significant differences observed. However, the transgenic calli expressing *PorbcL* exhibited a significant increase in *PorbcL* transcript levels compared to their WT and CK counterparts (Figure 6A).

Subsequent RT-qPCR analyses highlighted the significant up-regulation of specific genes in the *PorbcL*-overexpressing transgenic calli. Specifically, the expression of *PoSERK*, *PoAGL15*, and *PoGPT1* was markedly increased by 13-fold, 9-fold, and 233-fold, respectively (Figure 6B,D,E). Conversely, the expression of *PoLEC1* was down-regulated (Figure 6C). Notably, the empty vector pCAMBIA1302 alone did not result in significant changes to the transcript levels of these genes in the tree-peony calli.

### 2.7. Morphological and Anatomical Observations of Callus

Based on the characteristic differences in cell size, arrangement, and staining intensity between embryogenic callus cells (which tend to be small, densely arranged, and deeply stained) and non-embryogenic cells (which may exhibit larger cell size, loose arrangement, and lighter staining), observations and analyses of cell slices were conducted. This revealed that the *PorbcL*-overexpressing transgenic calli of tree peony exhibited a more compact cellular structure and a significantly larger embryogenic cell mass compared to both WT and CK calli after 25 days of culture (Figure 7A–C,*a*–*c*). This observation suggested that elevated *PorbcL* expression significantly facilitated the development of embryogenic cell clusters in the calli. Notably, the WT calli demonstrated a very low rate of embryogenesis, while there was no discernible difference in the number of embryogenic cell masses between the WT and CK calli. However, a significant increase in embryogenic cell masses was observed in the 35s::*PorbcL* transgenic calli (Figure 7D).

### 2.8. Endogenous Plant Hormone Contents

Since no anatomical variations were discernible between the WT and CK calli, CK, which shared the same transformation background as 35s::*PorbcL*, was designated as the control for subsequent experiments investigating the effects of *PorbcL* overexpression on the endogenous hormone levels in tree-peony calli. Notably, the overexpression of *PorbcL* exerted distinct effects on the concentrations of various endogenous phytohormones during the embryogenesis process in the tree-peony calli (Figure 8).

During embryogenic induction in tree-peony callus tissue, the overexpression of *PorbcL* led to a significant increase in endogenous IAA content, while significantly decreasing the levels of ABA and BAPR. Overexpression of *PorbcL* resulted in extremely low levels of ABA during the early stages of embryonic establishment, extremely high levels of IAA in the later stages of embryonic establishment, and extremely low levels of BAPR throughout the entire process of embryonic establishment. Thus, *PorbcL* played a regulatory role in the dynamic changes of IAA, ABA, and BARP levels during tree-peony embryogenesis.

Plant embryogenesis is a process that relies on the coordinated regulation of phytohormones. Therefore, the changes in the ratio of IAA to other phytohormones during tree-peony embryogenesis were determined (Figure 9). The results showed that the IAA/ABA, IAA/BAPR, IAA/ACC, IAA/5DS, and IAA/BL ratios in callus tissues were higher in the *PorbcL*-overexpressing group than in the control group. The maximum ratio of IAA/ABA was on the 10th day, whereas the maximum ratio of IAA to other phytohormones occurred at the late stage of embryonic establishment. These findings indicated that *PorbcL* overexpression played a regulatory role in the establishment of embryogenesis in tree-peony callus tissue by increasing the endogenous IAA content and its ratio to ABA, BAPR, ACC, 5DS, and BL.

## 3. Discussion

Somatic embryogenesis exploits the property of cellular totipotency, harnessing the potential of individual cells to develop into complete organisms. Notably, the *SERK* gene has emerged as a pivotal molecular marker during the nascent stages of plant embryogenic callus formation, as it displays a robust and specific expression pattern within this tissue. Overexpression of *SERK* has been demonstrated to significantly increase the occurrence of embryogenic calli, underscoring its role in promoting this critical process [42,43,44]. Genes are transcribed and translated into proteins. However, proteins rarely act independently in plant biological processes—they usually function as part of complexes [45]. By meticulously screening and validating interacting proteins, we can unlock novel functions of established proteins, discover previously unknown proteins and genes, unravel regulatory pathways, and ultimately forge innovative strategies to advance our understanding of plant biology [46,47,48]. In this study, we constructed yeast libraries of cDNA from tree-peony embryos at different stages of development for the first time. The capacity of the library was 6.2 × 10^6^ cfu/mL. The average length of randomly detected bands ranged from 500 bp to 750 bp, which met the criteria for library integrity and coverage. This provided a solid foundation for library screening.

In this study, *PoSERK* was used as the bait to screen the yeast library for interacting proteins, and 20 positive interactors were obtained. One of them was the Rubisco large subunit. Rubisco is a bifunctional enzyme in chloroplasts; it is not only the main enzyme for carbon absorption during photosynthesis but also the main regulator of photorespiration [41,49]. Usually, callus tissues are colorless or white because they lack chloroplasts and cannot photosynthesize [50]. However, some cells around the callus tissue contain chloroplasts—these cells are known as green tissue cells. Rubisco participates in the initial key step of carbon fixation in the Calvin cycle, converting free CO_2_ from the air into substances stored inside the plant. In this way, Rubisco supports photosynthesis by supplying energy and nutrients to callus tissue, enabling it to continue to grow and develop [41,49].

A previous study found that Rubisco was essential for embryo development in rapeseed. Rubisco catalyzes the carboxylation of ribulose bisphosphate to produce phosphoglycerate, converts carbohydrates into oils through glycolysis, and recovers the CO_2_ produced during this process, thereby improving carbon utilization efficiency [50]. The active sites of Rubisco are mainly located in the rbcL subunit, and the amino acid composition of this subunit largely determines its catalytic activity. Therefore, it is crucial to analyze the related functions of the *rbcL* gene [36,37]. In this study, the *PorbcL* gene was successfully cloned from tree peony. The bioinformatics analysis revealed that the 1761 bp ORF encodes a peptide comprising 586 amino acids. The PorbcL protein was anticipated to be an acidic, non-secretory, hydrophobic protein devoid of a transmembrane domain. Subcellular localization studies affirmed its presence within chloroplasts as well as at the plasma membrane. Furthermore, the interaction between PorbcL and PoSERK was reconfirmed by Y2H and BiFC analyses, and the interaction was observed to occur at the cell membrane and within the chloroplasts.

The specific location of gene products within cells is a prerequisite for their normal biological function. Previous studies have found that rbcL is enriched and expressed in embryos, where it participates in the carbon assimilation process of the Calvin cycle [51]. We observed in our study elevated transcript levels of *PorbcL* in both the roots and flowers of tree peony, with a gradual increase noted during embryonic development. The significant presence of *PorbcL* transcripts in the nutritional and reproductive organs/tissues of tree peony underscores its pivotal role in embryo growth and development, potentially contributing metabolic substrates and energy sources essential for embryogenesis.

Somatic embryogenesis is the reprogramming of cellular pluripotency, including the complex process of coordinating and precisely regulating the expression of genes related to somatic embryogenesis [31,52,53,54]. In this study, we overexpressed *PorbcL* in tree-peony callus to investigate its effect on the expression of key genes related to embryogenesis. We observed that the overexpression of *PorbcL* prominently up-regulated the expression of *PoSERK*, *PoAGL15*, and *PoGPT1*, all pivotal regulators in somatic embryogenesis. Furthermore, this overexpression markedly enhanced the quality of dense embryogenic cell clusters in tree-peony callus tissue, suggesting that *PorbcL* possesses the ability to augment somatic embryogenesis in tree peony.

In plants, endogenous phytohormones are chemical signals that regulate cell differentiation and development [54]. The induction and differentiation of somatic embryos depend on interactions among endogenous phytohormones such as auxin, cytokinin, and ABA [55,56,57,58]. Among them, auxin is crucial in the signal transduction pathway, leading to tissue generation and proliferation during somatic embryogenesis. Auxin can promote the expression of embryogenic marker genes such as *WUS, SERK,* and *BBM* in callus tissue [59,60,61]. In this study, we found that the overexpression of *PorbcL* significantly increased the IAA content in peony embryogenic callus tissue and significantly reduced the contents of ABA and BAPR. We detected significant correlations between the transcript level of *PorbcL* and the contents of endogenous IAA, ABA, and BAPR. Overexpression of *PorbcL* significantly affected the IAA/ABA, IAA/BAPR, IAA/ACC, IAA/5DS, and IAA/BL ratios in callus tissue, and the maximum values of these ratios occurred at the middle and late stages of embryogenic induction. These findings indicate that *PorbcL* participates in regulating the formation of embryogenic callus by increasing the content of endogenous IAA and modifying its ratio to other phytohormones.

## 4. Materials and Methods

### 4.1. Plant Materials and Treatment Methods

Embryos of tree peony (*Paeonia ostii* T. Hong et J. X. Zhang) were collected at 0, 5, 15, 25, 35, 45, 55, 60, 65, 75, 85, 90, 100, 110, and 130 days throughout the seed development process, from pollination to maturity. The seeds and flowers containing these embryos were collected from 10-year-old plants growing in sandy loam soil at Henan Agricultural University (Zhengzhou, China). The collected embryos were used as materials for yeast library construction. Simultaneous to embryo collection, flowers, leaves, roots, stems and fully mature seeds were collected from the tree-peony plants at the 5, 60, 65, 75, 90, and 110-day embryo-sampling time points. All the samples were promptly frozen in liquid nitrogen and subsequently stored at −80 °C until analysis (Thermo Forma 991, Thermo Fisher Scientific, Waltham, MA, USA). Each sample comprised 0.5 g tissue with three replicates. In addition, axillary buds were selected for callus induction and tissue culture. The specific culture method was as described by Song et al. [9].

### 4.2. Yeast Library Construction and Screening of PoSERK-Interacting Proteins

The embryos at different stages of development were ground into powder in a pre-cooled mortar with liquid nitrogen and then mixed evenly. The total RNA was extracted from the tree-peony materials using the RNAiso Plus Kit (TaKaRa, Dalian, China). The RNA concentration and quality were measured using an ultraviolet spectrophotometer and by gel electrophoresis. The mRNA was separated and purified using magnetic beads with the Magnosphere™ UltraPure mRNA Purification Kit (9186, TaKaRa), according to the manufacturer’s instructions. The primary and secondary libraries were constructed with the assistance of Shanghai Nuoji Biotechnology Co., Ltd. (Shanghai, China).

The open reading frame (ORF) of *PoSERK* (GenBank: KF876175, without the stop codon) was cloned into pBT3-STE to construct the yeast two-hybrid bait vector pBT3-STE-PoSERK (Appendix A). Then, the pBT3-STE-PoSERK vector was transformed into NMY51 yeast-competent cells for self-activation and functional verification. The pBT3-STE-PoSERK bait vector and the yeast two-hybrid secondary library plasmids were co-transformed into yeast NMY51-competent cells. The cells were spread onto SD-TL plates (SD medium lacking leucine and tryptophan), which were then incubated at 30 °C upside-down for 3 days. Single colonies of uniform morphology (1–2 mm) were selected and streaked onto SD-TLHA plates (SD medium lacking tryptophan, leucine, histidine, and adenine), which were then incubated upside-down at 30 °C for 2–3 days. Single colonies with uniform morphology (1–2 mm) were selected and streaked onto SD-TLHA+X-α-Gal plates, which were then incubated upside-down at 30 °C for 2–3 days. Then, blue-colored single colonies of uniform morphology (1–2 mm) were selected for PCR verification following the instructions provided by the Zhuangmeng Biology Yeast Positive Identification Kit (Beijing, China). The PCR products were sequenced by Sangon Biotech Co., Ltd. Blast comparisons and other analyses of the sequencing results were conducted at the NCBI database (https://www.ncbi.nlm.nih.gov). Protein–protein interaction (PPI) network analysis and Gene Ontology (GO) enrichment analysis were conducted using STRING 12.0 (https://cn.string-db.org/), and the interaction network diagram was constructed using Cytoscape 3.5.1.

### 4.3. Cloning and Analysis of PorbcL

Total RNA was extracted from 0.5 g tree-peony tissues (leaves (the third leaf from the apex), seeds and stems) using the RNAiso Plus Kit (TaKaRa). Each sample was prepared with three replicates. Then, cDNA was carried out employing the Prime Script RT Reagent Kit (TaKaRa). Specifically tailored primers (Appendix A) were created for the purpose of amplifying and cloning the entire coding sequence (CDS) of the *PorbcL* (GenBank: OR972713). The ORF of *PorbcL* was translated using BioXM2.7 software. The physicochemical properties of the putative protein and its conserved domains were determined using tools on the NCBI website (https://www.ncbi.nlm.nih.gov). The domain analysis was conducted using the online software SMART9 (http://smart.embl-heidelberg.de). DNAMAN8 and MEGA5.2 were used for multiple sequence alignment and phylogenetic tree construction, respectively.

### 4.4. Subcellular Localization Analysis

Utilizing gene-specific primers (Appendix A), the ORF of *PorbcL* was amplified and subsequently cloned into the pCAMBIA1302 vector, yielding the construct designated as 35S::*PorbcL*:GFP. This construct was introduced into *Agrobacterium tumefaciens* GV3101 and then transferred into leaf cells of *Nicotiana benthamiana* for subcellular localization analysis. The methods used for these procedures were described by Sparkes et al. [62]. The GFP fluorescence was visualized at a magnification of 40× by a confocal laser scanning microscope (A1+, Nikon, Tokyo, Japan). Quantitative analysis of co-localization was performed utilizing the Fiji software v2.3.0.

### 4.5. Verification of the Protein-Protein Interaction

Y2H Point-to-Point Verification: The ORF of *PorbcL* was amplified and subsequently cloned into the pPR3N vector to generate pPR3N-PorbcL with gene-specific primers (Appendix A). The pBT3-STE-PoSERK and pPR3N-PorbcL plasmids were co-transformed into yeast NMY51-competent cells, and the transformed cells were spread onto SD-TL and SD-TLHA plates. Then, 1–2 mm positive colonies were picked from the SD-TLHA plates and spread onto SD-TLHA+X-α-gal plates. After the colonies turned blue, individual colonies were collected and diluted 10-, 10^2^-, 10^3^-, and 10^4^-fold. The diluted samples were spotted onto SD-TLHA+X-α-gal plates and incubated at 30 °C for 3–6 days.

BiFC Verification: The ORFs of *PoSERK* and *PorbcL* were amplified and cloned into the YFP-N and YFP-C vectors with gene-specific primers (Appendix A), respectively. Each of the YFP-N-PoSERK and YFP-C-PorbcL plasmids was separately transformed into *A. tumefaciens* GV3101, and the transformed cells were spread onto LB+Kan+rif plates and incubated in the dark for 2–3 days. Then, single colonies 1–2 mm in diameter were selected for colony PCR identification. The YFP-N-PoSERK-GV3101 and YFP-C-PorbcL-GV3101 strains were cultured in LB+Kan+rif liquid medium with shaking until OD600 = 0.6–0.8. Each strain was then transferred to LB+Kan liquid medium and cultured with shaking to OD600 = 1.0. The YFP-N-PoSERK-GV3101 and YFP-C-PorbcL-GV3101 cultures were mixed in equal proportions, and then the mixture was injected into the abaxial epidermis of *N. benthamiana* leaves using a syringe. After 3 days, the leaves were observed under a confocal laser scanning microscope (A1+, Nikon).

### 4.6. Callus Transformation

The ORF of *PorbcL* was cloned into pCAMBIA1302 to construct the overexpression vector 35S::*PorbcL*, which was then transformed into *Agrobacterium tumefaciens* strain LBA4404. The pCAMBIA1302 vector, devoid of any insert, was individually introduced into *A. tumefaciens* cells to serve as the empty vector control (CK), while a non-transformed callus acted as the blank control (WT). *A. tumefaciens* harboring each construct was used to transform tree-peony calli, and the *A. tumefaciens* cells were removed before expression analysis. At 5 days after transformation, RT-qPCR analyses were conducted to detect and quantify *PorbcL* in the transformants, thereby proving that the callus was indeed transgenic. The absence of the *vir* gene in the *PorbcL*-containing calli was confirmed by gene sequencing analysis and gel electrophoresis, ensuring that the formed callus was not contaminated by *Agrobacterium tumefaciens*. The protocol used was described by Shen [63]. The callus was observed through paraffin sectioning, gene expression analysis, and hormone content measurement after 20 days of embryogenic induction culture, as described Song et al. [9]. Each sample was prepared with three replicates.

### 4.7. Hormone Content Determination

Samples of transgenic calli of CK and 35S::*PorbcL* were meticulously pulverized using liquid nitrogen, and the powder was transferred into 10 mL centrifuge tubes. The levels of endogenous plant hormones in the transformed calli were quantified using ultra-high-performance liquid chromatography (LC-30A, Shimadzu, Kyoto, Japan) coupled with mass spectrometry (Triple TOF 5600+, AB SCIEX, Foster City, CA, USA). Each sample was prepared with three replicates. The standards of plant hormones were acquired from Sigma-Aldrich (St. Louis, MO, USA). The extraction and quantification procedures followed the previously established protocols [22].

### 4.8. Morphological and Anatomical Observations

After culturing on a somatic embryogenesis induction medium for 20 days, the various calli (CK, WT, and 35S::*PorbcL*) were collected for morphological and anatomical observations. After fixation in formalin/acetic acid/alcohol = 1/1/18 (*v*/*v*/*v*), paraffin-embedded sections were observed under a light microscope. Five sections of each callus were selected for observation and analysis of cell structure. Based on the morphological and cellular structural characteristics of non-embryogenic and embryogenic calli, a statistical analysis was conducted to quantify the number of embryogenic cell clusters in the sections [6,64]. The sections (thickness, 10 μm) were prepared by Wuhan Service Biotechnology Co., Ltd. (Wuhan, China). Each callus type was analyzed in triplicate (three individual calli).

### 4.9. Real-Time Quantitative PCR Analysis

The transcript levels of the somatic embryogenesis-related genes *PoSERK* (GenBank: KF876175), *PoLEC1* (GenBank: ON454840), *PoAGL15* (GenBank: ON454841), and *PoGPT1* (GenBank: ON392713) were determined by RT-qPCR to verify the establishment of an embryogenic callus. The Primer-BLAST online tool (https://www.ncbi.nlm.nih.gov/tools/primer-blast/primertool.cgi, accessed on 10 August 2022) was used to design forward and reverse primers. The internal reference gene utilized was the *β*-tubulin gene (EF608942) of tree peony, detailed in Appendix A. For the analysis of gene transcript levels, the SYBR Premix Ex Taq™ II Kit (TaKaRa) was employed, adhering strictly to the manufacturer’s guidelines. A standardized two-step approach with three biological replicates was adopted, as outlined by Forootan and Kralik [65,66]. The 7500 Real-Time PCR System (Applied Biosystems, Foster City, CA, USA) facilitated the quantitative analysis, and the relative transcript levels were determined employing the ∆∆Ct method.

### 4.10. Statistical Analysis

SPSS 18.0 software (SPSS Inc., Chicago, IL, USA) was used to the analysis of variance (ANOVA). Duncan’s multiple range test was applied at a significance level of *p* ≤ 0.05 to identify significant differences among means.

## 5. Conclusions

For the first time, a Y2H library of cDNA from tree-peony embryos has been established at different stages of development. Employing *PoSERK*, a critical transcription factor in tree-peony somatic embryogenesis, as bait, we screened 20 positive interactors. Of these, 18 proteins had not been previously documented to interact in the model organism *Arabidopsis thaliana* (Figure 10A, excluding HSP90-5 and HSP90-7). Using the *Arabidopsis thaliana* database as a reference, we analyzed the protein interaction network of SERK, its interacting proteins, and key regulators of somatic embryogenesis. Gene Ontology (GO) enrichment analysis indicated that 5 genes were associated with somatic embryogenesis, 13 genes were associated with post-embryonic development, 12 genes were associated with auxin response, and 30 genes were associated with hormone response pathways (Figure 10B). These results significantly refined the interactive regulatory modules focused on key regulators of somatic embryogenesis, including *SERK1*, *AGL15*, *LEC1*, *LEC2*, and *ABI3*.

Further functional analysis was conducted on *PorbcL*, which is closely related to somatic embryogenesis, to elucidate its regulatory mechanism in tree-peony somatic embryogenesis. The results showed that PorbcL was predominantly localized to the chloroplast and plasma membrane. The interaction sites of PorbcL and PoSERK were localized at both the cell membrane and chloroplasts. *PorbcL* transcript levels were significantly higher in tree-peony roots and flowers compared to other plant tissues and gradually increased during embryonic establishment. Overexpression of *PorbcL* in the tree-peony callus directly or indirectly up-regulated the embryogenesis-related genes *PoSERK*, *PoAGL15*, and *PoGPT1* and down-regulated *PoLEC1*, thereby promoting the formation of embryogenic callus. Overexpression of *PorbcL* in the tree-peony callus led to dynamic changes in endogenous hormone contents (IAA, ABA, BAPR, 5DS, ACC, and BL), which regulated the early embryonic development of tree peony. Based on these results, we speculate that *PorbcL* coordinates endogenous phytohormones, resulting in an elevated IAA concentration and increased ratios of IAA to ABA, BAPR, ACC, 5DS, and BL. These changes collectively contribute to the reprogramming of somatic pluripotency and promote the establishment of embryogenesis, in which the elevated auxin content relative to ABA and cytokinin may be critical for somatic embryogenesis (Figure 11). Further research is necessary to elucidate the specific regulatory mechanisms involved in this process.

## Figures and Tables

**Figure 1 plants-13-02697-f001:**
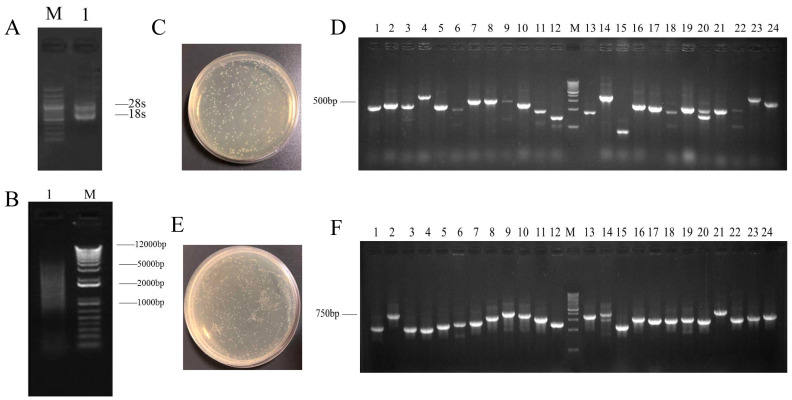
Yeast library construction ((**A**), Agarose gel electrophoresis of Total RNA (M, 250 bp NDA Ladder; 1, Total RNA); (**B**), Agarose gel electrophoresis of mRNA (M, 1 Kb Plus DNA Ladder; 1, mRNA); (**C**), Identification of primary library capacity; (**D**), Identification of inserted fragment length (M, 250 bp NDA Ladder; 1–24, Gene fragment); (**E**), Identification of secondary library capacity; (**F**), Identification of inserted fragment length (M, 250 bp NDA Ladder; 1–24, Gene fragment)).

**Figure 2 plants-13-02697-f002:**
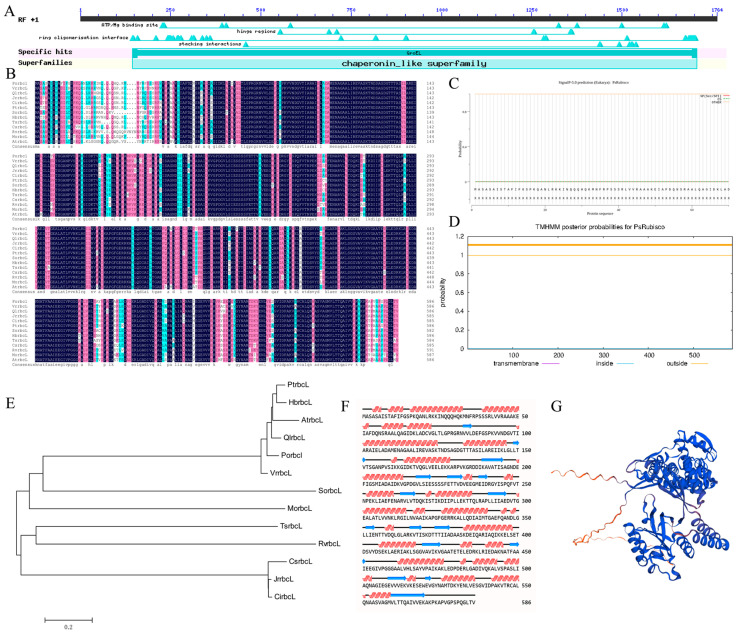
Sequence analysis of PorbcL ((**A**), Conserved domains; (**B**), Sequence alignment; (**C**), Signal peptide analysis; (**D**), Transmembrane domain analysis; (**E**), Phylogenetic analysis; (**F**), Secondary structure of protein; (**G**), Tertiary structure of protein).

**Figure 3 plants-13-02697-f003:**
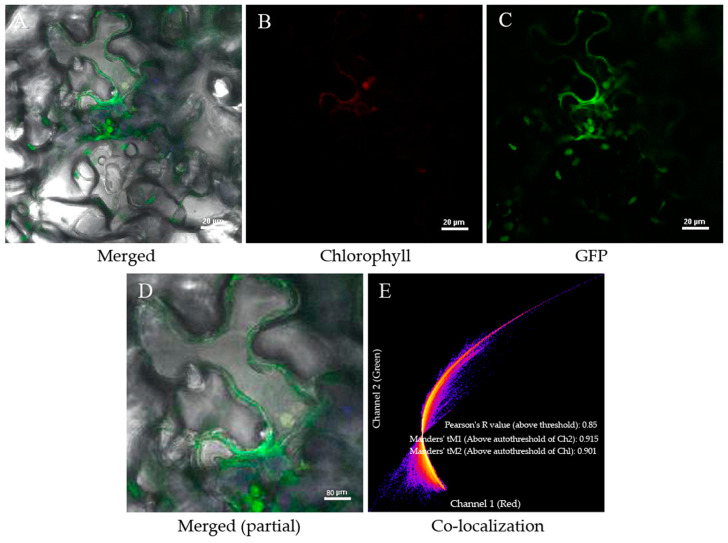
Subcellular localization of PorbcL ((**A**), Merged; (**B**), Chlorophyll; (**C**), GFP; (**D**), Merged (partial); (**E**), Co-localization). The location of the fusion protein is indicated by the presence of green fluorescence. Scale bar: (**A**–**C**) 20 μm, (**D**) 80 μm.

**Figure 4 plants-13-02697-f004:**
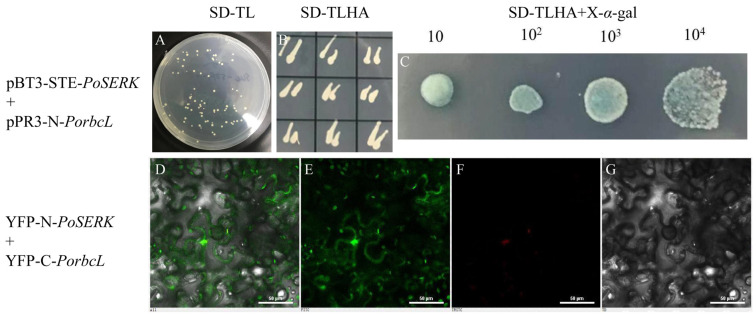
Interaction validation of *PoSERK* with *PorbcL* by Y2H and BiFC (Y2H: (**A**), SD-TL; (**B**), SD-TLHA; (**C**), SD-TLHA+X-*α*-gal, the bacterial solution was diluted 10, 10^2^, 10^3^, 10^4^ times respectively. BiFC: (**D**), Merged; (**E**), FITC (green); (**F**), TRITC (red); (**G**), Bright. Scale bar = 50 μm).

**Figure 5 plants-13-02697-f005:**
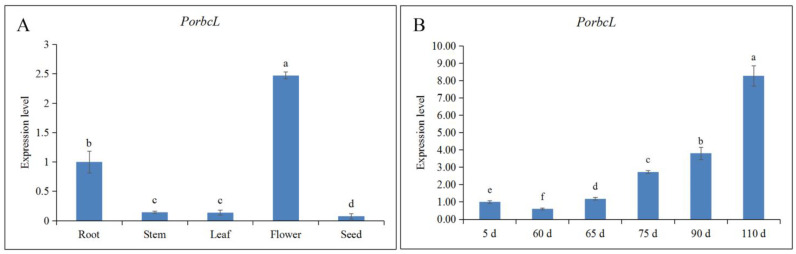
Spatiotemporal expression levels of *PorbcL* in tree peony ((**A**), Different parts; (**B**), Different zygotic embryo development stages). Standard error indicated by vertical bars (*n* = 3). Significant differences were denoted by distinct letters, as determined by the Duncan’s multiple range test (*p* ≤ 0.05).

**Figure 6 plants-13-02697-f006:**
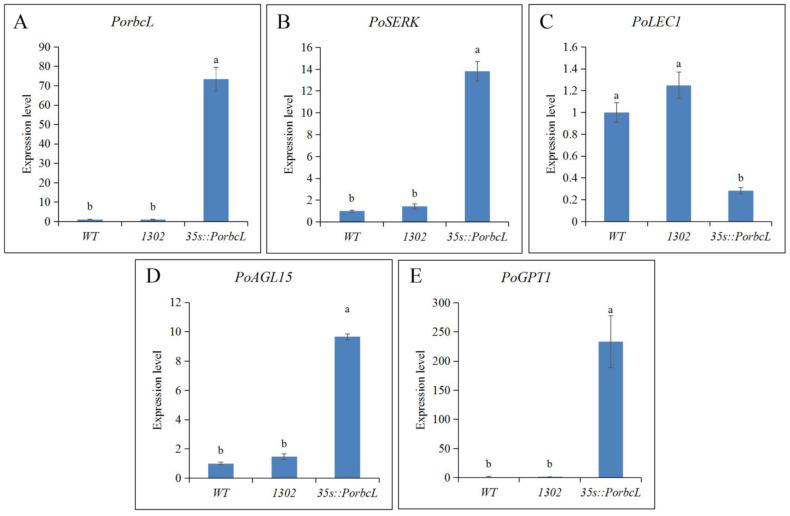
Positive identification and quantification of embryogenesis-related gene expression in *PorbcL* transgenic calli of tree peony via RT-qPCR analysis ((**A**), *PorbcL*; (**B**), *PoSERK*; (**C**), *PoLEC1*; (**D**), *PoAGL15*; (**E**), *PoGPT1*). Standard error indicated by vertical bars (*n* = 3). Significant differences were denoted by distinct letters, as determined by the Duncan’s multiple range test (*p* ≤ 0.05).

**Figure 7 plants-13-02697-f007:**
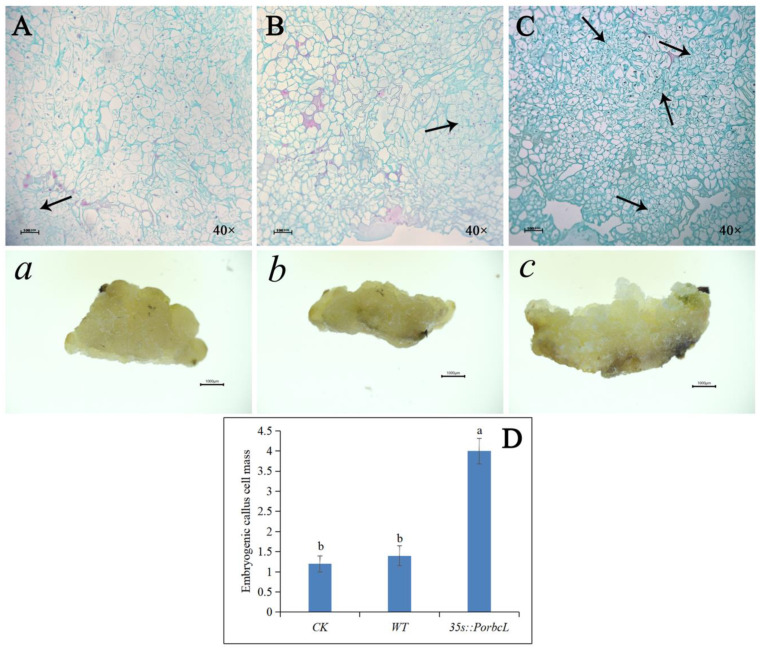
Morphological and histological observation of *PorbcL* transgenic callus ((**A**,***a***), WT: non-transformed callus; (**B**,***b***), CK: pCAMBIA1302; (**C**,***c***), *35s::PorbcL*; (**D**): Quantitative analysis of embryogenic callus cell mass. Standard error indicated by vertical bars (*n* = 3). Significant differences were denoted by distinct letters, as determined by the Duncan’s multiple range test (*p* ≤ 0.05). The black arrow points to the embryogenic callus mass. Scale bar: (**A**–**C**) 100 μm, (***a***–***c***) 1000 μm).

**Figure 8 plants-13-02697-f008:**
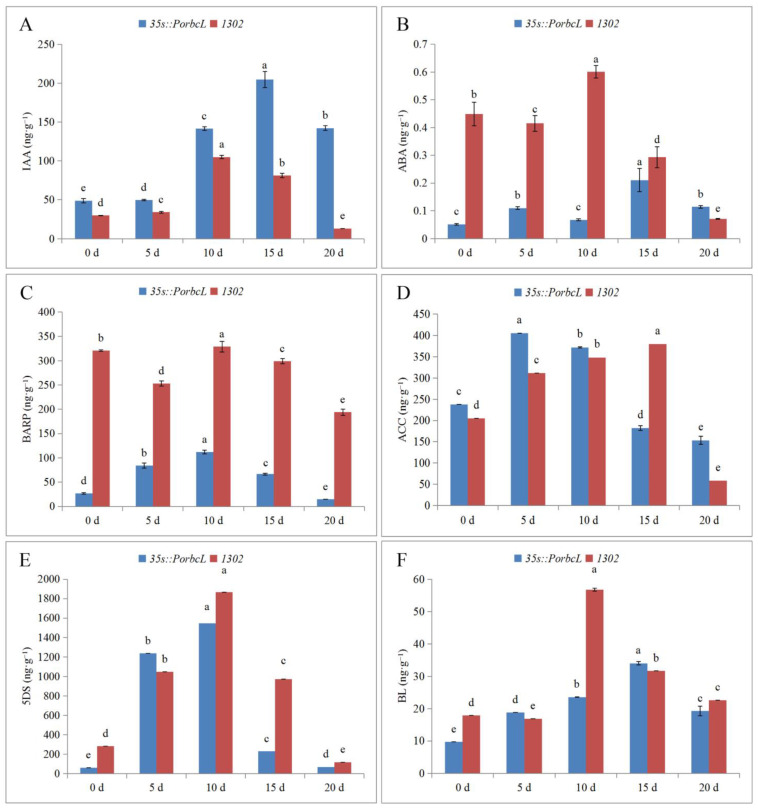
Endogenous hormone contents of *PorbcL* transgenic callus ((**A**), IAA; (**B**), ABA; (**C**), BARP; (**D**), ACC; (**E**), 5DS; (**F**), BL). Standard error indicated by vertical bars (*n* = 3). Significant differences were denoted by distinct letters, as determined by the Duncan’s multiple range test (*p* ≤ 0.05).

**Figure 9 plants-13-02697-f009:**
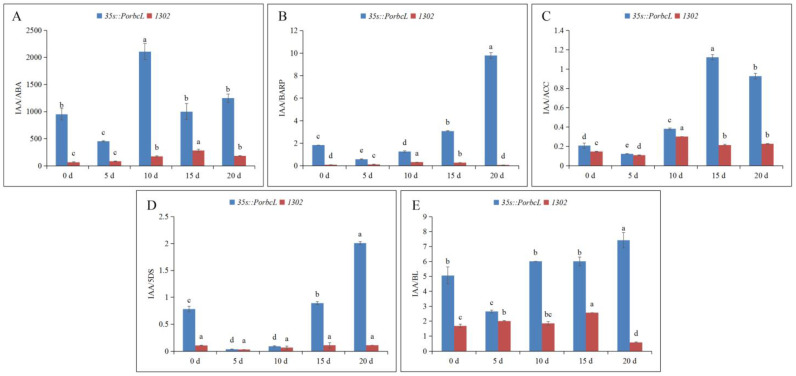
Endogenous hormone ratios of *PorbcL* transgenic callus ((**A**), IAA/ABA; (**B**), IAA/BARP; (**C**), IAA/ACC; (**D**), IAA/5DS; (**E**), IAA/BL). Standard error indicated by vertical bars (*n* = 3). Significant differences were denoted by distinct letters, as determined by the Duncan’s multiple range test (*p* ≤ 0.05).

**Figure 10 plants-13-02697-f010:**
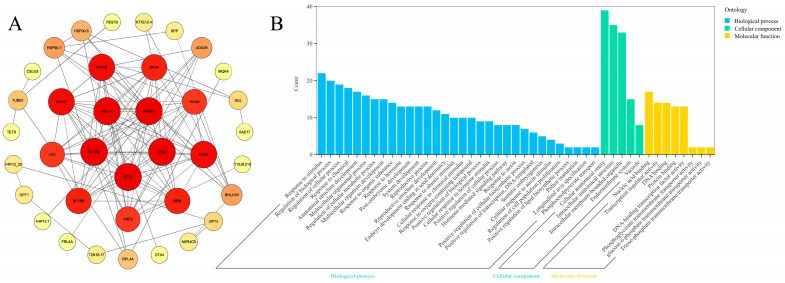
Interaction network diagram and enrichment analysis of SERK and its interacting proteins ((**A**), protein–protein interaction (PPI) network, where the diameter of the circle’s ranges from large to small and the color gradient transitions from deep to light, representing the degree values from high to low; (**B**), Gene Ontology (GO) enrichment analysis).

**Figure 11 plants-13-02697-f011:**
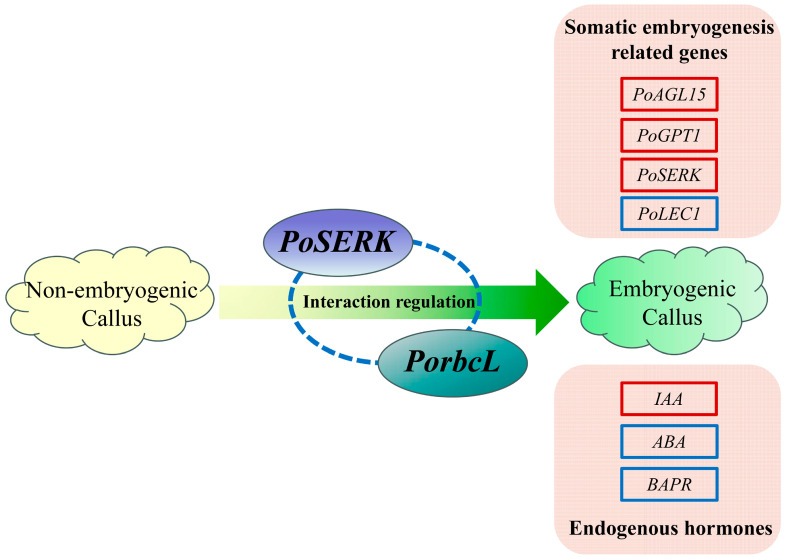
Model map of *PorbcL* promotion of embryogenic callus formation in tree peony. The red frame represents up-regulation; the blue frame represents down-regulation.

**Table 1 plants-13-02697-t001:** Positive interaction sequence alignment analysis.

Number	Name	Homologous Sequence (*Arabidopsis thaliana*)	Biological Function
1	Protein GPR107-like	AT1G10980.1	Transmembrane receptor family protein
2	Glucose-6-phosphatephosphate translocator (GPT1)	AT1G61800.1	Dynamic domestication of photosynthesis
3	Post-GPI attachment to proteins factor 3(PGAP3)	AT2G46710	Regulation of the periodic changes in secondary cell wall pits
4	Kelch repeat-containing protein At3g27220-like	At3g27220	Galactose oxidase/Kelch repeat superfamily protein
5	Transcriptional adapter ADA2-like	AT4G16420	Controls cell proliferation, mediates auxin and cytokinin signaling, and may be involved in freezing tolerance pathways
6	Golgin candidate 4	AT2G46180	Connects vesicles to the Golgi membrane, maintaining the overall structure of the Golgi apparatus
7	Probable cysteine protease RD21B	AT5G43060	Involved in phagocytosis and clearance of excess intracellular material
8	Endoplasmin homolog	AT4G24190	Regulates meristem formation by modulating the folding of CLV proteins
9	60S ribosomal protein L4	AT3G09630	Structural component of ribosomes
10	FRIGIDA-like protein 4a	AT3G22440	Cell differentiation, flower development
11	Uncharacterized GPI-anchored protein At4g28100	AT4G28100	Unknown
12	Signal peptide peptidase	AT2G03120	Intramembrane-cleaving aspartic protease, catalyzes intramembrane proteolysis of signal peptides, and is involved in reproductive tissue development
13	Probable acyl-activating enzyme 17, peroxisomal	AT5G23050	Forms acetyl-CoA to activate carboxylic acids
14	Tetraspanin-8	AT2G23810	Involved in the regulation of cell differentiation
15	Rubisco large subunit protein (rbcL)	ATCG00490	Involved in the carboxylation of ribulose 1,5-bisphosphate, carbon dioxide fixation, and photorespiration
16	Universal stress protein A-like protein	AT3G01520	Binds amide nitrogen and carbonyl oxygen
17	Tubulin beta-1 chain	AT1G75780	Microtubule cytoskeleton organization, originates from the mitotic cell cycle
18	Transcription factor bHLH35	AT5G57150	Transcriptional regulation, embryonic development, flower development, senescence, etc.
19	NADPH-dependent aldo	AT2G37770	Catalyzes the reduction of saturated and α,β-unsaturated aldehydes
20	Heat shock protein 90-5, chloroplastic	AT2G04030	Chloroplast biogenesis and maintenance

## Data Availability

The original contributions presented in the study are included in the article/Appendix A, further inquiries can be directed to the corresponding authors.

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
