# Peer review of "Functional Analysis of the PoSERK-Interacting Protein PorbcL in the Embryogenic Callus Formation of Tree Peony (*Paeonia ostii* T. Hong et J. X. Zhang)"

_plants, 2024, doi:10.3390/plants13192697_

Round 1

Reviewer 1 Report

Comments and Suggestions for Authors

The peer-reviewed manuscript "Functional Analysis of the PoSERK-Interacting Protein PorbcL in Embryogenic Callus Formation of Tree Peony (Paeonia ostii)" devoted to fundamental problem such as role of PorbcL gene for induction of embryogenic callus in tree peony. The possibility of results applying of this research also highlights the potential of future publication. However, the article has a number of inaccuracies and questions, which are presented below.

1)     Add of the author-classifier to the genus and species names plants. Additionally, for the first time in the text, the generic name should be presented in full, while further in the text it should be abbreviated. Please carefully check the correct Latin names throughout the text.

2)     Be careful when abbreviations are first mentioned. Abbreviations at the first mention should be written in full and decoded. For example, L32 (but decreased contents of abscisic acid and 6-benzyladenosine).

3)     Citation in the text of the manuscript is not carried out according to the rules of the Plants journal. Please read carefully and format the references in the manuscript in accordance with the rules of journal.

4)     Fig 1D, F. Tracks 1-24 need to be aligned

5)     Fig 7. It is not clear how it was carried out of quantitative analysis of embryogenic callus cell mass?

6)     In the Materials and Methods section, how many plants were used in each biological replicate? How many analytical replicates were there for the biochemical assessment? How many control and treated plants were used for RNA isolation for each experiment? Leaves of what tier and age were selected for expression analysis?

7)     How did the authors prove that the callus was transgenic? How did the authors prove that the formed callus was not contaminated with Agrobacterium?

Reviewer 2 Report

Comments and Suggestions for Authors

Dear Authors,

The report by Yinglong Song et al entitled: “Functional Analysis of the PoSERK-Interacting Protein PorbcL  in Embryogenic Callus Formation of Tree Peony (Paeonia ostii)” shows the correlation of specific proteins and specific plant growth regulators (especially IAA) in Paeonia ostii embryogenic callus induction and maintaining.

This studies provide interesting and clear data, of practical importance, which can be used in cultivation of Peony tree. The work is broad and well documented and I think it will interest many readers, however, a few minor revisions (language and editorial) must be carried out before this manuscript is acceptable for publication.

I recommend to publish this article in “Plants” after correction of some minor problems/errors:

Abstract:

Line 18-19: “The lack of a mature embryo regeneration system for tree peony (Paeonia ostii) limits its industrial development” – in my opinion this sentence does not bring nothing it suggest that this species has the reproductive problems, if this is true, Authors should to explain it more clear.

Line 23: “… tree peony embryos” – do you mean “somatic embryos”? it should be clarified

Line 33: “increased ratio of IAA to other hormones” – it is better to specify the PGRs (plant growth regulators) and do not use the term “hormones” but PGRs or phytohormones.

Keywords:

Should be in alphabetical order and more specifically to the aim of work

Introduction:

Line 42-43: information about the peony should be supplemented, there are several reports about the medical use of oils, and other secondary metabolites of this beautiful plant.

Line 63-65 : “Because this breeding method relies on a mature tissue culture regeneration system, it is crucial to construct a somatic embryo regeneration system for tree peony.” – I do not understand why it so important and why the regeneration system could not be based on the organogenesis? If the Authors consider genetic stability in this context it should be clarified.

Line 66: “The tree peony tissue culture regeneration system is not yet mature” – “mature” is not suitable for this reason, please verify is.

Line 73: “embryonic”, “embryonic” – should not be “embryogenic”?

Line 97-107: all this information presented here are of result and conclusion meaning but here is the place to put the research objective, the aim of the experiments, please correct it.

Results:

Line 248-264: Morphological and Anatomical Observations – in this chapter there is lack of more particular cytological information for example, the good marker of morphogenic potential is the presence of extra cellular matrix (ECM), did you observe it?  I would question the comparability of the callus mass (Quantitative analysis of embryogenic callus cell mass), there is no information about the size of the calluses clamps taken for measurement. I did not find information about the standardization of these activities. Please complete this.

Discussion:

Line 375-376: “increased ratio of IAA to other hormones” – it is better to specify the PGRs (plant growth regulators) and do not use the term “hormones” but PGRs or phytohormones.

Materials and Methods:

Line 480-487: Morphological and Anatomical Observations – like in Line 248-264 (Results)

Figure 10. embryogenic callus or embryonic callus – should be verified

Comments on the Quality of English Language

Dear Authors,

the manuscript needs slight English correction.

Round 2

Reviewer 1 Report

Comments and Suggestions for Authors

Dear authors,

Thank you for the clarifications that have been included in the revised version of the manuscript. The resubmitted article can be accepted for publication in the Plants journal.

Author Response

Thank you so much for your affirmation and support of our work. We will continue to contribute our manuscripts in the future research!